# LEARNING BY DISTILLING CONTEXT

## ABSTRACT

Language models significantly benefit from context tokens, such as prompts or scratchpads. They perform better when prompted with informative instructions, and they acquire new reasoning capabilities by generating a scratch-pad before predicting the final answers. However, they do not *internalize* these performance gains, which disappear when the context tokens are gone. Our work proposes to apply context distillation so that a language model can improve itself by internalizing these gains. Concretely, given a synthetic unlabeled input for the target task, we condition the model on "[instructions] + [task-input]" to predict "[scratch-pad] + [final answer]"; then we fine-tune the same model to predict its own "[final answer]" conditioned on the "[task-input]", without seeing the "[instructions]" or using the "[scratch-pad]".

We show that context distillation is a general method to train language models, and it can effectively internalize 3 types of training signals. First, it can internalize abstract task instructions along with explanations, so we can recursively update the model parameters with new instructions and overwrite old ones. Second, it can internalize concrete training examples, and it outperforms directly learning with gradient descent by 9% on the SPIDER Text-to-SQL dataset; furthermore, combining multiple context distillation operations can internalize more training examples than what the context window size allows. Finally, we show preliminary evidence that it can internalize step-by-step reasoning on 8-digit addition, and such a newly acquired capability proves to be useful for other downstream tasks.

## 1 INTRODUCTION

Recent work has shown that language models significantly benefit from context tokens. When prompted with task definitions, language models can perform zero-shot learning (Wei et al., 2022a; Sanh et al., 2022), and the performance further improves with additional in-context examples and explanations (Chen et al., 2022; Scheurer et al., 2022). They also acquire the capability to perform more complex tasks by generating step-by-step reasoning in the context window before predicting the final answer (Nye et al., 2021b; Wei et al., 2022b; Zhou et al., 2022).

However, language models cannot *internalize* these performance gains, which disappear when the context tokens are gone. Consequently, we always need to pay extra computation for running inference on context tokens; this is undesirable, as sometimes the task instructions and the scratch-pad can be more than 10x longer than the actual task inputs. Furthermore, it is unclear how to leverage the context tokens when their total length exceeds the context window size. These shortcomings are analogous to how humans are slow at performing complex cognitive tasks (Wason & Evans, 1974) and can hold only a limited amount of information in the working memory (Baddeley, 1992).

Humans get around this by practicing. Consider, for example, learning to type your friends' phone numbers. The first few times you type it, you need to consciously recall the number using working memory and slowly decide which button to press. After repeatedly typing the same number, it becomes a habit and you can type the number quickly without conscious reasoning. Through repeated practice, the knowledge of your friend's phone number is "distilled" into your muscle memories.[1] This mechanism for distilling knowledge is critical for learning complex tasks because it allows us

---

[1]See declarative learning vs. procedural learning for a friendly but more in-depth discussion. `https://en.wikipedia.org/wiki/Declarative_learning`

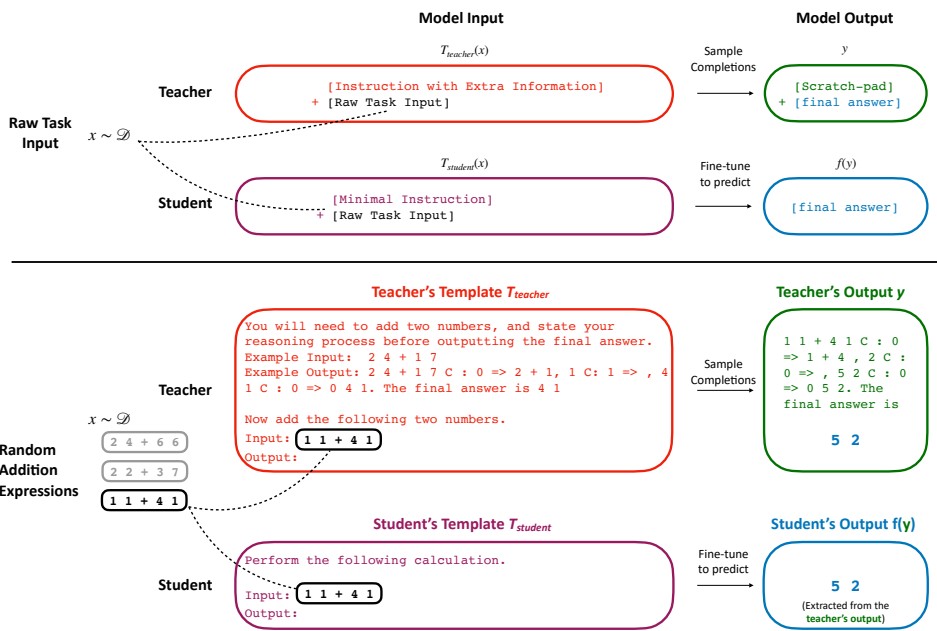

Figure 1: **Top.** An overview of our context distillation framework. We sample a raw task input, form the teacher's prompt by pre-prending a detailed instruction that might contain more examples and explanations, and ask the language model to conditionally sample a scratch-pad and a final answer. Then we fine-tune the same language model to directly predict the final answer with a minimal instruction. We formalize this framework mathematically in Section 2. **Bottom.** An instantiation of our framework that internalizes step-by-step reasoning for 8-digit addition.

to incrementally build up our knowledge and skills, so that we can learn to accomplish increasingly complex tasks.

We propose to apply a similar method, context distillation, to fine-tune language models. For example, as shown in Figure 1, to make language models internalize the step-by-step addition capability, we first synthesize a large number of "practice" addition questions; we then ask the model to follow the more informative instruction to reason step-by-step before generating the target answer; finally, we fine-tune the language model to directly predict the answer conditioned on a simpler student prompt. As a result, by practicing on a lot of addition problems, the ability to add is distilled into its parameters. We formally state our generalized context distillation framework in Section 2.

Section 3 shows that we can apply context distillation to a wide range of settings: learning from abstract statements, learning from concrete examples, and learning from step-by-step reasoning. Section 3.1 (Figure 2) shows that context distillation can effectively internalize task instructions along with natural language explanations from Natural-Instructions-V2 (Wang et al., 2022b); additionally, we can teach the student to associate numerical indices with certain tasks, and then we can recursively re-assign these task indices, overwriting the student's past associations. Section 3.2 (Figure 3) shows that context distillation can be used to internalize Text-to-SQL training examples from the SPIDER dataset (Yu et al., 2018) into Incoder (Fried et al., 2022), and it outperforms directly learning with gradient descent by 9% for 8-shot adaptation; additionally, we show that as we distill more training examples than can fit in the context window, we observe continual improvements in performance. Section 3.3 (Figure 3) shows that we can internalize step-by-step reasoning to perform 8-digit addition, and such a capability can transfer to downstream question answering tasks; we hope this preliminary results can generalize to more complex and realistic tasks that larger models can perform with chain-of-thoughts reasoning (Wei et al., 2022b; Zhou et al., 2022).

Overall, context distillation demonstrates promising potential as a general method for learning. As discussed in Section 4, we predict that future models will be better able to learn from context than today's models, and researchers will use these models to tackle increasingly complex tasks that

require more background knowledge and longer reasoning chains. Therefore, we anticipate our method to be increasingly useful in the future.

## 2    CONTEXT DISTILLATION

We introduce the main components and the intuition of our context distillation framework in Section 2.1, describe our algorithm for single round distillation in Section 2.2, explain how to distill multiple contexts recursively or simultaneously in Section 2.3, and describe various implementation details to make it efficient and stable in Section 2.4.

### 2.1    INTUITION AND MAIN COMPONENTS

We explain our method by contrasting it with the classical distillation methods (Hinton et al., 2015a). These classical methods ask the teacher model with parameter $\theta_{\text{TEACHER}}$ to generate a label $y$ for a given input $x$, and train the student $\theta_{\text{STUDENT}}$ to mimic the teacher by predicting $y$ conditioned on $x$. Typically, $\theta_{\text{TEACHER}} \neq \theta_{\text{STUDENT}}$ when the algorithm starts, and the distillation process is driven by the difference between their parameters. In contrast, under context distillation, $\theta_{\text{TEACHER}} = \theta_{\text{STUDENT}}$ when the training starts, and the distillation process is instead driven by the differences in the $x$ and $y$ that they see and predict.

To design such a difference that drives the distillation process, our framework requires the model developers to provide four components: a raw task input distribution $\mathcal{D}$, a teacher template $T_{\text{TEACHER}}$, a student template $T_{\text{STUDENT}}$, and an answer extractor $f$. We introduce them below.

**Raw Task Input Distribution $\mathcal{D}$.**   $\mathcal{D}$ is a distribution of strings, which are typically the "core" inputs of the target task of interest. For example, if the target task is to classify movie review sentiment, the input distribution could be defined as random movie reviews. Generally, there are many ways to define a raw task input distribution: we can use rule-based method to generate random strings, sample from a pool of unlabeled data, or conditionally sample from a language model. We explicitly distinguish raw task inputs from the whole "input prompt" to the language model, which will be obtained after applying the templates below.

Except for the synthetic experiments in Section 3.3, all experiments in our paper create $\mathcal{D}$ via few-shot prompting and do not assume access to a pool of unlabeled data.

**Teacher Template $T_{\text{TEACHER}}$.**   $T_{\text{TEACHER}}$ is a mapping from strings to strings, which transforms raw task inputs to the input prompts for the teacher model. As shown in Figure 1, the teacher template usually contains more detailed instructions, explanations, and examples about the task.

**Student Template $T_{\text{STUDENT}}$.**   $T_{\text{STUDENT}}$ is a mapping from strings to strings, which transforms raw task inputs to the input prompts for the student model. As shown in Figure 1, this template usually still contains minimal information about the task so that the request in the prompt is not under-specified. However, compared to the teacher prompt, it incorporates far fewer explanations and training examples of the task, and such a difference transfers this useful context information into the student parameters.

**Answer Extractor $f$.**   $f$ is a mapping from token sequences to token sequences, which extracts the final answer (a sub-sequence of tokens) from the full teachers' generation. As shown in Figure 1, $f$ strips away the intermediate reasoning process, and the students need to internalize the step-by-step reasoning process to directly predict what the teacher predicts at the end.

We now describe context distillation formally using the mathematical terms we just introduced.

### 2.2    FORMAL DESCRIPTION

Our algorithm first samples an $x$ from $\mathcal{D}$ and ask the language model to sample a completion $y$ conditioned on $T_{\text{TEACHER}}(x)$; we then fine-tune the language model to predict $f(y)$ conditioned on $T_{\text{STUDENT}}(x)$. Throughout the distillation process, $\theta_{\text{TEACHER}}$ is fixed.

Formally, let $\theta_{\text{STUDENT}}$ and $\theta_{\text{TEACHER}}$ be the parameters of a language model, and define $P_\theta(\cdot|\text{PROMPT})$ to be the probability distribution of the completions conditioned on the prompt. We optimize

$$\mathcal{L}_{\mathcal{D}, T_{\text{STUDENT}}, T_{\text{TEACHER}}, f, \theta_{\text{TEACHER}}}(\theta_{\text{STUDENT}}) = \mathbb{E}_{x \sim \mathcal{D}}[\mathbb{E}_{y \sim P_{\theta_{\text{TEACHER}}}(\cdot|T_{\text{TEACHER}})}[\log P_{\theta_{\text{STUDENT}}}(f(y)|T_{\text{STUDENT}}(x))]] \tag{1}$$

Notice that the definition of $\mathcal{L}$ depends on five variables $\mathcal{D}, T_{\text{STUDENT}}, T_{\text{TEACHER}}, f$, and $\theta_{\text{TEACHER}}$. To keep the notation uncluttered, we will only include the necessary subscripts if the rest can be inferred from the surrounding text.

## 2.3 COMBINING MULTIPLE UPDATES

We now introduce simultaneous distillation and recursive distillation, two straightforward variants that combine multiple context distillation operations, allowing us to internalize ensembles or sequences of contexts, enabling a form of learning that is not possible with just a single prompt.

**Simultaneous Distillation.** To simultaneously perform $K$ different context distillation operations represented by $\mathcal{D}_{1...K}, T_{\text{TEACHER/STUDENT}, 1...K}, f_{1...K}$, we can optimize the total loss:

$$\mathcal{L}_{\text{TOTAL}} := \sum_{k=1}^{K} \mathcal{L}_{\mathcal{D}_k, T_{\text{STUDENT}, k}, T_{\text{TEACHER}, k}, f_k} \tag{2}$$

Simultaneous distillation is especially useful when the prompts contain independent instructions and in-context training examples, but their total length exceeds the language model context window size.

**Recursive Distillation.** To perform $R$ rounds of context distillation operations recursively, we can inductively define $\theta_0$ as the initial language model, and $\theta_{k+1}$ to be the parameters after fine-tuning with the loss function

$$\mathcal{L}_r := \mathcal{L}_{\mathcal{D}_r, T_{\text{STUDENT}, r}, T_{\text{TEACHER}, r}, f_r}, \tag{3}$$

with $\theta_r$ as the initialization. Essentially, for each round $r$ of context distillation we can obtain a new student, which then becomes the teacher and the student's initialization for the next round. This allows a language model to recursively update itself via context distillation.[2]

## 2.4 IMPLEMENTATION DETAILS

A naive method to optimize Equation 1 is to sample $y$ and directly fine-tune the student to predict the hard label of $f(y)$: such a method wastes the token logit information and results in noisy gradients. Instead, we minimize the token-level KL divergence between the student and the teacher (i.e., learn from the soft labels). However, this leads to another issue: the vocabulary space of language models is often on the order of 50k, and the full soft labels consume as much as 200KB memory per decoding step. Therefore, we approximate the it with an empirical distribution of 100 token samples. Such a technique saves memory by a factor of 500. Appendix 8.1 shows more details.

## 3 EXPERIMENTS

**Overview.** We apply context distillation to three types of settings. Section 3.1 applies context distillation to internalize abstract instructions and natural language explanations; additionally, language models can recursively update themselves via context distillation to associate each task instruction with a task id. Section 3.2 applies context distillation to internalize concrete training examples, and it can outperform directly learning with gradient descent on the SPIDER Text-to-SQL dataset; additionally, we show that simultaneous distillation can be used to internalize more training examples than the context window can fit. In Section 3.3, we apply context distillation to internalize the ability to perform 8-digit addition step-by-step; additionally, such a capability can transfer to other downstream question-answering tasks even when the scratch-pad is not present. In all cases, the student's input length is significantly shorter than the teacher's, and we may reduce the context length by as much as 11 times, saving us a large amount of compute at inference time.

---

[2]Choi et al. (2022) first proposed this recursive formulation but only evaluated it qualitatively.

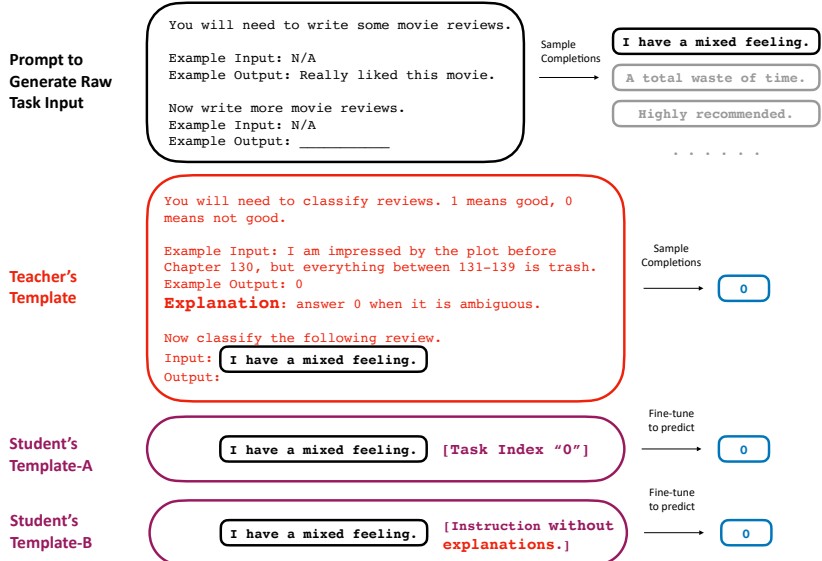

Figure 2: We use context distillation to internalize abstract task instructions and associate each of them with a task id. For example, after distilling the context with the teacher's template and student template-A, the student should perform sentiment classification whenever it sees the index "[0]". We perform an additional ablation study by using the student template-B without the explanation to verify that context distillation can be used to learn from natural language explanations. The raw task inputs are sampled from the same teacher model via few-shot prompting (top).

**Baselines and Oracles.** At the beginning of the distillation, the teacher and the student networks have identical parameter values. However, the teacher has an advantage because it has access to additional context while the student does not. The student's goal is to mimic the teacher to internalize the teacher's context. Therefore, the student's performance needs to improve after context distillation to support the claim that context distillation is successful. The teacher's performance is generally an upper bound on the student's performance.

**Naming Conventions.** We use the following names when reporting the results.

- "Teacher" = **initial** language model's (LM) performance **with** additional context.
- "Pre-distill Student" = the **initial** LM's performance **without** additional context.
- "Post-distill Student" = the **distilled** LM's performance **without** additional context.

Note that whether a system is called teacher or student is not based on its parameter values (e.g., initial vs. distilled) but on whether it uses context.

### 3.1 LEARNING FROM ABSTRACT INSTRUCTIONS AND EXPLANATIONS

We apply context distillation to internalize abstract instructions. In all experiments, unless mentioned otherwise, the answer extractor is the identity function and we use few-shot prompting to sample raw task inputs (see Figure 2).

**Dataset and Language Models.** We use Natural-Instructions-V2 for all the experiments on this section. Natural Instructions is a dataset of 1600+ diverse language tasks, where each task is associated with a natural language task description, input-output examples, and explanations about why certain outputs are correct or incorrect (Wang et al., 2022b). We trained our teacher language model (TK-Instruct) by fine-tuning LM-adapted T5-11B (Raffel et al., 2020) on its training set, and the details can be seen in Appendix 8.5. For evaluation, we use their dataset's official metric, Rouge-L, to calculate the performance averaged across the 10 tasks we selected.

**Design Choice.**   We select 5 tasks from their evaluation split where the teacher can most significantly improve the performance when prompted with extra natural language explanations compared to task description only, and 5 where they improve the least. We chose these tasks because they also allow us to investigate how well context distillation can internalize natural language explanations in Appendix 8.2 under different teacher's performance.

**Hypothesis 1: context distillation can internalize abstract task instructions.** To test this hypothesis, we defined the student template as the identity mapping, i.e., the student only sees the raw task input, while the teacher template contains the task instruction, which consists of a task description, 2 positive in-context examples, and 2-negative in-context examples (Figure 2 teacher's template). The teacher's performance is 43.4 Rouge-L, establishing an upper bound for the student. Before context distillation, the student's performance is 9.0, since it does not know what task it should perform. After context distillation, the student's performance significantly increases to 34.7. Context distillation successfully internalized abstract task instructions. Finally, we used 11.1 times fewer inference time tokens when evaluating the student verses the teacher.

In light of emerging trends to learn from from natural language explanations  (Scheurer et al., 2022; Lampinen et al., 2022), we present experiments on distilling in-context explanations in Appendix8.2.

**Hypothesis 2: recursive distillation can overwrite past updates.**  We study recursive distillation using four classification tasks, superglue_copa_text_completion, meta_woz_task_classification, tweetqa_classification and rocstories_title_classification. We define a new task id association challenge: each task is associated with an index, and when the student model sees an index, it needs to perform the corresponding task without looking at its instruction; we train the student model to do this via simultaneous context distillation, where the student sees the task index while the teacher sees the task instruction (Figure 2 student template A). After we use context distillation to train the student to associate each task-id with the corresponding instruction, we shuffle the task-id association, perform context distillation again, and then evaluate the model's performance on the newly defined association to measure the ability of context distillation to overwrite previous updates.

We define two metrics for this task id association challenge. First, we will measure the average accuracy (correct association accuracy) of the student on the four tasks when they are prompted with the corresponding index. Second, it is plausible that the student can learn to associate the task input distributions, rather than the index, with the corresponding instructions. For example, suppose that id "[0]" corresponds to classifying sentiment and the task input distribution is movie reviews, while "[1]" corresponds to whether it is sports related and the raw input distribution is news articles, then the student might cheat by always classifying sentiment whenever it sees a movie review, regardless of what id it sees. Therefore, we also measure the average accuracy when we prompt the model with the wrong task id, and we want this number to be low (wrong association accuracy): for example, if the model sees "[0]" and a news articles, it should have low accuracy at classifying whether it corresponds to sports, because "[0]" corresponds to sentiment classification.

We experimented with two variants of simultaneous distillation: 1) the "naïve" one, where each $\mathcal{D}_k$ contains only the input distribution for one single task, and 2) the "mixed" one, where we define each $\mathcal{D}_k$ as a mixture of all the raw task input distributions. As shown in Table 1, under the "naïve" input distribution, the student model can cheat by associating the task input distribution, rather than the task id, with the task it needs to perform; on the other hand, the "mixed" variant successfully over-writes the past task id association.

Can the language model recursively update itself based on new task-id associations? It is not obvious that it would work, since it is plausible that the model forgets how to follow the instruction properly after being fine-tuned to associate task with Ids. We recursively apply context distillation for three rounds and find in Table 1 right that the performance does not drop. Along with findings from (Ramasesh et al., 2022), these results suggest that catastrophic forgetting might pose little trouble to recursively applying context distillation on larger instruction-tuned models.

## 3.2   Learning from Concrete Examples

We show that context distillation can internalize training examples, and therefore can be a potential alternative to directly learning from these examples with gradient descent; additionally, simultaneous distillation allows the model to learn from more in-context examples when their total length exceeds

Table 1: We apply context distillation to train the student to associate task instructions with task ids, and then recursively apply context distillation repeatedly to overwrite the previous association. We report the correct association accuracy ("correct") and the wrong association accuracy score ("wrong") over all four tasks we associate. For all entries we report the 95% confidence interval. **Left**: Performance significantly improves after context distillation, and mixed distillation successfully prevents the model from cheating by memorizing the input distribution. **Right**: Task-id association performance does not drop after several rounds of recursive updates.

| model | correct ↑ | wrong ↓ |
|---|---|---|
| Teacher | $81 \pm 4$ | - |
| Pre-distill Student | $49 \pm 5$ | $48 \pm 5$ |
| "Naïve" Post-distill Student ($r = 2$) | $\mathbf{68} \pm 5$ | $61 \pm 5$ |
| "Mixed" Post-distill Student ($r = 2$) | $66 \pm 5$ | $\mathbf{16} \pm 4$ |

| model | correct ↑ |
|---|---|
| Round $r = 1$ | $68 \pm 5$ |
| Round $r = 2$ | $66 \pm 5$ |
| Round $r = 3$ | $68 \pm 5$ |

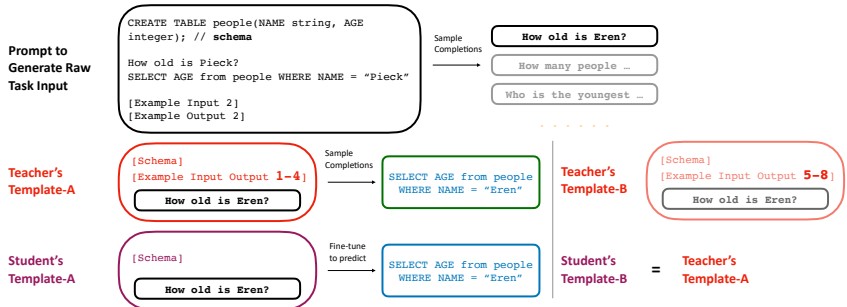

Figure 3: The teacher template A contains additional training examples 1-4 compared to the student template A, and context-distilling them outperforms directly learning from them with gradient descent. Additionally, by simultaneously applying the teacher's templates A and B (Section 2.3), more in-context examples (5-8) can be distilled, even though their total length might exceed the context window size. The raw task inputs are sampled from the same teacher model via few-shot prompting (top).

the context window length. In all experiments, the answer extractor is the identity function and we use few-shot prompting to sample raw task inputs (Figure 3).

**Dataset and Language Models.** We use the SPIDER text-to-SQL dataset (Yu et al., 2018) for our experiments. Each database in SPIDER is associated with a schema and a list of English questions annotated with SQL query. The task is to predict the SQL query given the schema and the question. For the teacher language model, we chose Incoder-6.7B (Fried et al., 2022), which is pre-trained on code. For each experiment, we randomly sampled eight text-to-SQL pairs for each database as in-context examples and evaluate on the rest by calculating the exact set match accuracy.

**Design Choice.** We chose this task because prior work (Rajkumar et al., 2022) has demonstrated that SOTA models are able to effectively utilize in-context examples for this task; therefore, our approach is more likely to outperform gradient descent. More generally, in-context learning performance improves when we scale up (Chen et al., 2022; Brown et al., 2020), and hence we conjecture that context distillation can outperform gradient descent on more tasks for future models.

**Hypothesis 3: context distillation sometimes outperform directly learning with gradient descent.** We define the student template to be a database schema followed directly by the question, whereas the teacher template contains the database schema followed by four in-context examples (see Figure 3 teacher's template-A). As we see in Table 2, context distillation outperforms learning via gradient descent on four examples by 8.6% in exact-set match accuracy, a margin which further increases by 0.4% when training on eight examples. Therefore, context distillation can be used as an alternative to directly learning from training examples with gradient descent.

| Model | 4 Examples | 8 Examples |
|---|---|---|
| Teacher | 27.7 | 28.2 |
| Pre-distill Student | 0.3 | 0.3 |
| Post-distill Student | **22.1** | **27.9** |
| Direct Gradient Descent | 13.4 | 18.9 |

Table 2: Comparing context distillation to gradient descent on the SPIDER text-to-SQL validation set. We see that context distillation outperforms directly learning via gradient descent on four examples by 8.6% (exact set match); this margin further increases when learning from eight examples.

**Hypothesis 4: context distillation enables learning from more training examples when their total length exceeds the context window size.** For this experiment, we select four databases from the SPIDER training set which have particularly long database schema, such that it is possible to fit four training examples into Incoder's context window but not eight. Therefore, we include 4 training examples in the student template (Figure 3, student's template B.), which would lead to the best in-context learning performance ex-ante given the context window size. To internalize more training examples, we perform simultaneous distillation and sample teacher templates by including a random subset of four in-context examples out of eight (Figure 3, teacher's template A and B). After context distillation, our student achieves an exact set match of $16.2 \pm 0.6$, which improves over the pre-distillation student performance of $14.22 \pm 0.8$, hence confirming our hypothesis.

### 3.3 PRELIMINARY EVIDENCE FOR LEARNING FROM STEP-BY-STEP REASONING

We show that context distillation can effectively internalize a skills acquired by generating step-by-step reasoning, and such a skill can benefit the model for other downstream tasks. Unless otherwise mentioned, in this section we use an $f$ that will extract the final answer from the teacher's full output, as shown in Figure 1 bottom.

**Dataset.** We define the input distribution $\mathcal{D}$ to be uniform over all possible 1 through 8 digit addition questions, identical to those used by Nye et al. (2021a). We report addition accuracy for each experiment. We hope that these results can generalize to more complex and realistic tasks that larger models can perform with chain-of-thoughts reasoning (Wei et al., 2022b; Zhou et al., 2022).

**Hypothesis 5: context distillation can internalize step-by-step reasoning.** To test this hypothesis, we obtain a teacher model that can use scratch-pad to perform step-by-step reasoning by fine-tuning the LM-adapted T5-small on a dataset of 500 addition expressions, where the model needs to first generate a scratch-pad before predicting the final answer. We then perform context-distillation with an $f$ that extracts the final answer from the teacher's output as shown in Figure 1 bottom. As shown in Table 3, after distillation, the ability of the student to perform **direct** additions (without using scratch-pad) improves from 0% to 94.7%, implying that context distillation internalizes step-by-step reasoning.

We compare this to several other transfer learning and multi-task learning baselines that use the same amount of training data in Table 3. Under transfer learning, we first fine-tune the student model to predict the scratch pad, and then fine-tune it to directly predict the final answer. Under multi-task learning, we fine-tune the model to predict scratch-pad and the final answer independently. Both variants perform significantly worse ($> 20\%$) than context distillation. In addition to the gains in reasoning ability over baselines, we also saved inference time compute in our evaluations: specifically we used 8.0 times fewer tokens at inference time when evaluating the student compared to the teacher.

**Hypothesis 6: the reasoning cability internalized by scratchpad distillation can transfer to other related reasoning tasks.** To test this hypothesis, we distill the addition scratch-pads on our TK-Instruct model, and evaluate capability transfer to other tasks. We obtain the teacher language model by fine-tuning TK-Instruct on a distribution of both natural instructions data and 500 addition scratchpad examples (see Appendix 8.9). For the context distillation training, we define the student template to be a description of the addition task followed by two direct answer in-context examples, and similarly the teacher template contains the task description and two in-context examples

| | Teach | Pre-Dist | Post-Dist | Sc→Dir | Sc+Dir |
|---|---|---|---|---|---|
| 8 Digit Addition Accuracy % | 93 | 0 | **95** | 72 | 61 |

Table 3: Distilling addition scratchpads on T5-small. "Teach" refers to the teacher LM's performance using scratch-pad. "Pre-Dist" refers to the student's performance before distillation; "Post-Dist" refers to the student's performance of direct addition (without scratch-pad) after distillation;"Sc→Dir"/"Sc+Dir" refers to our transfer/multi-task learning baseline. Context Distillation performs the best for direct addition.

with scratch-pad answer. To prevent the student from catastrophically forgetting its in-context learning ability during distillation, we mix our 10k distillation data-points with a distribution of 65536 randomly selected examples from Natural Instruction-V2.

The student's accuracy on directly answering 8-digit addition questions improves from 1% to 17%, while the student's performance on Natural Instructions remains roughly the same (from RougeL 57 before distillation to 58 after), implying that the student did not lose its original capabilities to follow instructions. Additionally, we evaluate the student's performance on a set of related reasoning tasks. We then use a template to synthesize simple questions that require knowledge to add two numbers, for example, "A has 7 turkies. B has 2 turkies. How many turkies do they have altogether?". On this synthetic dataset, the student's performance significantly increases from 17% to 30% after context distillation, implying that the capability to perform direct addition can transfer to other related applications.

## 4 RELATED WORK

**Prompting and Instruction Tuning.** Many recent works show that language models can learn from abstract task definitions (Zhong et al., 2021; Mishra et al., 2022; Wei et al., 2022a), natural language explanations (Scheurer et al., 2022; Wang et al., 2022b), and concrete in-context examples (Min et al., 2022; Chen et al., 2022). We anticipate the in-context learning performance to improve further in the future (Kaplan et al., 2020), thus increasing the upper bound of what context distillation can achieve.

**Scratch Pad.** Many recent works show that language models perform better when it is required to generate a chain of reasoning steps before outputting the final answer (Zhou et al., 2021; Nye et al., 2021b; Cobbe et al., 2021; Wei et al., 2022b; Zhou et al., 2022; Lewkowycz et al., 2022). We anticipate context distillation to be increasingly useful, as the research community starts to tackle more difficult problems, which require more sophisticated skills and have longer problem descriptions and reasoning chains.

**Distillation.** There has been a large literature on distilling knowledge in a neural network (Hinton et al., 2015b; Adriana et al., 2015; Liu et al., 2019; Yang et al., 2020; Xie et al., 2020). Most related to our work, different sub-variants of context distillation have been independently discovered by different researchers. Wang et al. (2021) emphasizes the aspect of creating a dataset without any human annotations and uses a language model to generate task inputs and their labels. Choi et al. (2022); Askell et al. (2021) formulated the method of context distillation (also referred to as prompt injection), which distills a fixed input prompt; their method is a special case of our framework with an identity student template and an identity output selector. Additionally, they focused more on the benefit of saving computational resources, while we considered it as a general learning method.

## 5 CONCLUSION

We present context distillation as a general learning method, which can internalize abstract statements, concrete examples, and step-by-step reasoning. Given that 1) it is general and delivers strong performance, 2) future models will have stronger in-context learning capability, and 3) future tasks will have longer descriptions and reasoning chains, we anticipate our methods to be increasingly useful in the future.

## 6 ETHICS STATEMENT

Our method is still far from perfect for internalizing context information and should not be applied to high stake scenarios.

## 7 REPRODUCIBILITY

We include more experiment details in the appendix and release code for reproducing all experiments.

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

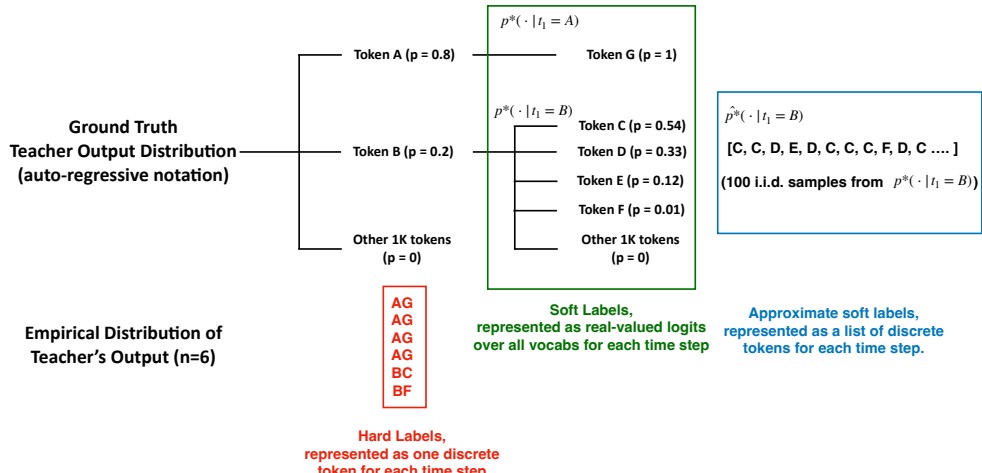

Figure 4: A concrete example used to illustrate how we performed context distillation by approximately optimizing token level KL divergence. The red ones are hard labels, which are noisy. The green ones are soft labels that are less noisy, but consumes a lot of memory. Therefore, we create a list of samples from the soft labels distribution, and use the empirical distribution to approximate the soft label distribution.

## 8 APPENDIX

### 8.1 OPTIMIZING APPROXIMATE TOKEN-LEVEL KL DIVERGENCE

We illustrate the intuition in greater details using a concrete example shown in Figure 4. The ground truth teacher's distribution over all output sequence is shown at the top. Ideally, we want the student's output token distribution $p_\theta$ to match the teacher's distribution $p^*$ (also referred to as the "ground truth" in this section); however, optimizing this object exactly requires enumerating all possible output sequences, which is computationally inefficient. Therefore, we sample $n$ (=6 in the figure, 100 in the actual experiments) output sequences from the teacher output distribution, and our goal is to use these output samples and the logit information to approximate $p^*$ and use it to efficiently train $p_\theta$.

First, naïvely training the student to predict the empirical distribution (red) is noisy and sample inefficient, since it might not contain lower probability sequences, e.g., the sequence BD, which occurs only 2% of the time. It wastes the logit information the teacher has provided us. Therefore, we should optimize

$$\mathcal{L}(\theta) = \mathcal{L}_{\text{1st Token}}(\theta) + \mathcal{L}_{\text{2nd Token}}(\theta),$$

where

$$\mathcal{L}_{\text{1st Token}}(\theta) = KL(p^*(t_1)||p_\theta(t_1)),$$

and

$$\mathcal{L}_{\text{2nd Token}}(\theta) = \frac{4}{6}KL(p^*(t_2|t_1 = A)||p_\theta(t_2|t_1 = A)) + \frac{2}{6}KL(p^*(t_2|t_1 = B)||p_\theta(t_2|t_1 = B)),$$

(4)

and we got the coefficient $4/6$ for the $KL$ term for $t_1 = A$ because 4 out of 6 the samples in the empirical distribution starts with token $A$.

Since our hardware does not have enough memory to hold two large models at the same time, we need to compute the teacher's logits on a large set of inputs, store them, and then load the student model, load the teacher's logits, and train the student accordingly. However, exactly computing $KL(p^*(\cdot|t_1 = B)||p_\theta(\cdot|t_1 = B))$ requires the full information of the probability distribution of $p^*(\cdot|t_1 = B)$, which requires 200 KB in memory[3]. Therefore, the teacher's logit will incur a large storage, which makes it impossible to fit them on the specialized hardware (i.e., GPU or TPU);

---

[3]Using float32 representation and Incoder tokenizer

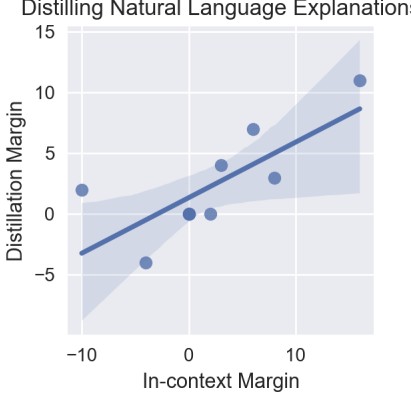

Figure 5: Distilling explanations for 10 tasks from the Natural Instructions V2 test set. Each point corresponds to a task in the figure. "In-context Margin" quantifies how much the explanations help the teacher by measuring the difference in RougeL of TK-Instruct with and without explanations on each task, and "Distillation Margin" quantifies how much the student learns from the teacher by measuring the difference between the RougeL score of the student after distillation and the RougeL score of TK-Instruct without explanations on each task. We observe a positive correlation between the utility of the explanations to the teacher and how much our student learns.

consequently, we might need to store these information on Secondary memory, which incurs a large communication cost. To compress the information of $(p^*(\cdot|t_1 = B)$ to fit on our specialized hardware, we need to use a *compact* and *unbiased* representation; therefore, we approximate $(p^*(\cdot|t_1 = B)$ with an empirical distribution of 100 sample tokens from this distribution (as shown in $\hat{p^*}$ Figure 4), which now only takes 0.4KB[4].

## 8.2 LEARNING FROM NATURAL LANGUAGE EXPLANATIONS

**Hypothesis 7: context distillation can learn from natural language explanations when they benefit the teacher.** In this experiment, we use the same teacher template as the experiment above; in contrast, in this experiment, we define a student template that is exactly the same as the teacher, but without explanations (Figure 2 student template B). We run context distillation to internalize the effect of natural language explanations, using the task's training split as $\mathcal{D}$.

Ideally, we want the student to fully internalize the effect of natural language explanations and match the teacher's performance. To measure how much the student internalizes, we define two quantities for each task: 1) the in-context margin, which measures the performance increase after we add the explanations to the context, and 2) the distillation margin, which measures the performance increase after we perform context distillation. We plot the margins for each task in Figure 5, and we observe a positive and statistically significant correlation ($r = 0.75, p = 1\%$). Therefore, context distillation can learn from natural language explanations when they benefit the teacher.

Notice that for our current model, not all tasks benefit from internalizing natural language explanations. However, we anticipate that future language models be more capable of responding to natural language explanations (Kaplan et al., 2020; Scheurer et al., 2022), hence improving the performance of context distillation.

## 8.3 CONTROLLED TEXT GENERATION WITH CONTEXT DISTILLATION

By distilling prompts that describe desired behavior (e.g. don't output toxic language or only generate positive sentiment text), we can instill a form of control in language models. To test this, we prompt our TK-Instruct model to complete negative movie reviews from the IMDB sentiment classification dataset (Maas et al., 2011).

---

[4]100 int32

|  | teacher | | | student w/o PI | | | student w PI | | |
|---|---|---|---|---|---|---|---|---|---|
|  | sent | RougeL | ent | sent | RougeL | ent | sent | RougeL | ent |
| positive | **94.0** | **4.8** | **13.0** | 0.24 | 0.6 | 2.5 | **0.96** | 4.8 | **14.4** |
| neutral | 0.54 | 3.6 | 9.9 | 0.24 | 0.6 | 2.5 | 0.34 | **7.0** | 14.2 |

Table 4: Controlled generation via-context distillation. The "sent" column reports the average sentiment score of generations from the model, and the "ent" column refers to the model's estimated output entropy in nats. We see that By injecting positive control directions into the model, we can obtain a model which outputs greater postive sentiment text without significantly sacrificing output coherence (RougeL) or output diversity (entropy).

The teacher template contains directions to complete the movie review specifically to end the review on a positive note, and the student template contains directions just to complete the review without any other specification. After distillation, the student should learn to internalize the abstract control preference that we expressed to the teacher model. We evaluate our models by querying GPT-3 for a sentiment classification on the model's output. We generate questions using a few-shot prompt to TK-instruct, and take 64 distillation steps with batch size 16 and AdamW optimizer with 1e-5 learning rate and 0 weight decay.

We see in Table 4, that by applying this procedure we are able to successfully control our language model.

## 8.4 FACTUAL KNOWLEDGE EDITING WITH CONTEXT DISTILLATION

Context distillation also enables a very natural way to edit the factual knowledge implicitly internalized by language models, by distilling a prompt that states a new or edited declarative fact. This is in contrast to prior works (Mitchell et al., 2021; De Cao et al., 2021; Meng et al., 2022) on fact editing, which instead perform a constrained optimization procedure that directly learns an edit to the model parameters, corresponding to the factual knowledge update.

We use the challenging Counterfact dataset from Meng et al. (Meng et al., 2022) to test our method's fact editing ability. Each instance of the Counterfact task involves editing the object of a given factual relation. Ideally the language model's should consistently apply the new fact under significant paraphrases to the original relation and context changes; the model should also not edit unrelated knowledge. To measure this, the Counterfact dataset provides a set of paraphrase prompts (significant paraphrases of the original fact, which the LM should consistently edit), neighborhood prompts (un-related facts that share the same object as the original pre-edit fact, which should not change), and attribute prompts (un-related facts which share the same object ad the new post-edit fact, which should not change).

We perform fact editing experiments on our TK-Instruct model. For a randomly selected fact corresponding to each of the 34 unique relations in the Counterfact dataset, we synthesize a teacher template, which includes a description of the fact to be edited and instructions to not edit unrelated facts. We also use GPT-3 to help us generate a new attribute, paraphrase, and neighborhood prompt for each fact edit to use as demonstrations of desired behavior in the prompt, alongside a natural language explanation of why the fact edit is or is not applied in each case. We generate the inputs $P(x)$ using a few-shot prompt to TK-Instruct.

In Table 5, we evaluate our model on the set of paraphrase and neighborhood prompts in the dataset. We report both the average score – $\mathbb{1}[P(\text{correct object}) > P(\text{incorrect object})]$ – and the average magnitude – $P(\text{correct object}) - P(\text{incorrect object})$ – under the language model, where $P(\text{correct object})$ and $P(\text{incorrect object})$ are the probability of the correct and incorrect objects under the model, when conditioned on the relevant subject and relation. We see that context distillation is largely able to recover the fact editing performance of the teacher, and preforms comparably in absolute score to current SOTA approaches to fact editing – ROME (Meng et al., 2022) and MEND (Mitchell et al., 2021) – even though our method makes slightly different assumptions and therefore isn't directly comparable (i.e. our prompt contains more information than these methods use).

| method | paraphrase | | neighborhood | |
|---|---|---|---|---|
| | score | magnitude | score | magnitude |
| Teacher | 73 | 29 | 58 | 8 |
| Pre-distill student | 34 | -3 | 80 | 4 |
| Post-distill student | 79 | 28 | 48 | -2 |
| GPT-3 | 65 | 3 | 75 | 17 |
| MEND | 65 | 12 | 38 | -12 |
| ROME | 89 | 33 | 74 | 4 |

Table 5: Performance of context distillation on fact editing. We see that context distillation is able to recover the paraphrase score of the teacher, but slightly under-performs in neighborhood score.

However, our prompted TK-Instruct teacher model generally performs poorly on neighborhood prompts, which also carries over to our distilled model. We expect this issue to be largely resolved by context distilling larger and more capable language models. To demonstrate this, we evaluate GPT-3 on this task with the same prompt, which we can see in Table 5 performs much better on these neighborhood prompts. While we cannot perform context distillation on GPT-3 due to limitations in OpenAI's API, we expect these improvements to also carry over to the distilled model.

## 8.5 INSTRUCTION-TUNED T5-11B.

Following the procedure of (Wang et al., 2022a), we fine-tuned the 11B T5 LM-adapted model (Raffel et al., 2019), on Natural Instructions V2 (Wang et al., 2022a), a large dataset of 1600+ language tasks which includes, for each task, a task description, positive and negative in-context examples, and natural language explanations of why the output is right or wrong for each of the in-context examples. Prior work (Wang et al., 2022a) has used this dataset to train instruction-tuned models on prompts consisting of only 2-positive examples or only 2-positive and negative examples with explanations. To maximize the flexibility of our instruction-tuned model, we instead instruction-tuned on a distribution of randomized prompts, which consist of randomly chosen 0 to 3 positive examples, 0 to 3 negative examples, and whether there is an explanation or not. We trained the model for 9728 steps with a batch size of 16 and AdamW optimizer on 32 TPU-V3 cores. The model achieves a RougeL score of $58$ on the 2 positive, 2 negative with explanation test split, of unseen natural instructions tasks.

## 8.6 FINETUNING LANGUAGE MODEL IMPLEMENTATION DETAILS

We run all experiments on 32 TPU-V3 cores, with the model parameters and optimizer states for fine-tuning sharded equally across all cores. Our codebase is built in Jax (Bradbury et al., 2018; Sabne, 2020) using the PJIT function to handle the model parallel training and inference.

## 8.7 GENERAL EXPERIMENT DETAILS

For all experiments, except where denoted otherwise, we distill on 4096 examples for 1 epoch with batch size 16 with AdamW optimizer. We use a learning rate of 1e-5 with TK-Instruct and 1e-4 with Incoder. We use 0 weight decay for all experiments.

## 8.8 DISTILLING CONCRETE EXAMPLES DETAILS

**Gradient descent details (hypothesis 4).** For gradient descent we fit all training examples into a single batch and train for a 25 epochs using the AdamW optimizer with a learning rate of 1e-5. We report the performance of the epoch with the highest average exact set match score across all databases.

**Distilling long contexts details (hypothesis 5).** To estimate the per-token log-probabilities of the $y$ sampled from the teacher ensemble for distillation, we average the probabilities of each $y$ under 8 teacher prompts. We estimate the teacher performance by performing greedy decoding on the ensemble of 8 prompts uniformly sampled from the set of all 4 choose 8 teacher prompts. For each

database, we distill two students with different in-context examples in the prompt, and we report the average exact-set match accuracy for both of these students.

## 8.9    DISTILLING STEP-BY-STEP REASONING DETAILS

**Distilling scratchpads with T5-small (hypothesis 6).**    All baselines were trained for 1000 epochs – except "Scratchpad then Direct" which was trained for 1000 epochs to predict scratchpads and then 1000 epochs to directly predict the answer – with a batch size of 8, a learning rate of 3e-4, and AdamW optimizer. We report performance at the end of training on 10k unseen addition problems.

**Transfering step-by-step reasoning details (hypothesis 7).**    Since TK-Instruct cannot successfully do scratchpad addition from a few shot prompt, to initialize the teacher, we first fine-tune TK-Instruct on the same distribution of 500 scratchpad examples from the previous experiment mixed in with 4096 randomly selected examples from the "2 positive" split of Natural Instructions-V2 training set, such that the teacher doesn't lose its ability to respond to prompts. We train the teacher for 32 epochs, at which point it achieves 97% accuracy on 2000 held-out scratchpad addition problems.

