# OpenReview forum: "Learning by Distilling Context"
_ICLR.cc/2023/Conference — Submitted to ICLR 2023_

### Official Review · Reviewer_Krod · 2022-10-23

**Confidence:** 4
**Correctness:** 2
**Technical Novelty And Significance:** 3
**Empirical Novelty And Significance:** 2
**Recommendation:** 3

**Clarity, Quality, Novelty And Reproducibility:**

* The core idea of the paper is very clear and well-written.
* Some experimental details are not super clear from the main paper. For instance, just to clarify:
    * Is it correct that the teacher model and the student model have the same size?
    * Is it correct that the training tasks and test tasks are mutually exclusive when instruction-tuned, and all models (including all baselines and the proposed model) are only using the labeled data on training tasks but not the test tasks?
    * And is it correct that the proposed model performs distillation on the test dataset? In other words, the proposed model uses the unlabeled datasets of the test datasets, whereas the baselines do not?


**Strength And Weaknesses:**

**Strengths**
* The paper tackles an interesting problem – building a model that learns from a concise instruction and performs the task as much as the model given a detailed instruction and examples does. The problem is well-motivated – reading a long instruction and examples is often very expensive and inefficient, and is even strictly bounded by the maximum sequence length of the Transformer.
* The idea of distilling from the model reading a long instruction to the model reading a shorter instruction is new.
* The paper specifies a series of hypotheses and shows empirical results that verify each hypothesis very clearly.

**Weaknesses**
* Overclaiming:
    * The paper made a strong emphasis that the proposed model is able to “internalize” the task, and made this claim many times overall in the paper. However, to my point of view, “internalizing the task” is exactly “fine-tuning”, and in fact, the proposed method is all about “better fine-tuning through distillation”. In this regard, I think the paper is overall overclaiming.
    * Some details that indicate the proposed model is based on more assumptions than the baselines are not explicitly mentioned. For instance, based on my understanding, the proposed model is trained on the unlabeled portion of the test datasets, whereas baselines are not. Is it correct? (I am not super sure about this and asked this question in the “clarity” section.) If it is true, it is a fairly strong assumption to assum the unlabeled data of the test datasets, and should be mentioned very clearly. Not making this clear seems overclaiming.
* Missing the most critical baseline
    * I think the most important baseline is a student model that is instruction-tuned on a particular template (either teacher’s template or student’s template) with no distillation. To my understanding, this baseline is not provided, except for one experiment (Table 2). Instead, the paper claims that the proposed model effectively learns from the teacher model because it is better than the pre-distill student model. However, to my understanding, the pre-distill model is not instruct-tuned, thus it is unclear if this gain is coming from instruction tuning, or distillation from a teacher model. For this reason, I strongly believe that the paper should have mainly compared with the model instruction-tuned but without distillation, and should not claim the effectiveness of the distillation method without this baseline.
    * Side note: I think “Direct Gradient Descent” is basically the model that is instruction-tuned but without distillation. Is this correct? I think the name is not very intuitive. To my understanding, it is using the template for the teacher model (long instruction, in-context examples) – and if it is correct, I think it’s also good to include the model using the template for the student model (concise instruction). This will support the assumption in the paper that longer instruction is more helpful than concise instruction, which this paper actually never explicitly showed (nor cited relevant prior work).
* Experiments are not very consistent with no specific reason specified. For instance:
    * In the experiments with Natural Instruction v2, the result is evaluated only on 5 datasets with the largest gains from instruction tuning and other 5 datasets with the smallest gains from instruction tuning. It is not clear why the paper chooses to do so instead of reporting all datasets, especially because the paper did not report these two sets of 5 datasets separately and discuss how their results differ.

Minor:
* While the paper claims that prior work in distillation is mostly focusing on reducing the model size, I don’t think it is true – previous work has shown that distillation helps even when the teacher model and the student model have the same size of the model.


**Summary Of The Paper:**

This paper introduces a new distillation method, with a teacher model given a long, detailed instruction as well as in-context examples, and a student model given a concise instruction. The motivation is that, while the model performs the task well by reading a detailed instruction and examples, it is often expensive at inference time, because it has a significantly longer context. Therefore, having a student model that operates with shorter instruction but works as well as the model given a long instruction and in-context examples is beneficial. The paper evaluates the proposed model on a range of scenarios, effectively showing that the student model effectively learns from the teacher model with long instruction or explanation, and outperforms baselines such as naive instruction tuning without distillation.

**Summary Of The Review:**

Overall, the paper is well-written, and the idea of dilstilling from the model reading detailed instruction to the model reading more concise instruction is interesting and new. However, I am concerned with overclaiming in the paper, missing critical baselines and inconsistency in evaluation of the model.

---

> ### Author Response · Authors · 2022-11-17
> **Response to Reviewer Krod**
>
> TL;DR:
> - **Our original draft compared the context-distilled models to the critical baselines you recommended.**
> - **We did not use any unlabeled data from the test distribution.**
>
> We appreciate your critical feedback! We guess there might be some serious miscommunications, so we improved our draft by adding more information at the beginning of the experiment section. In our paper,
>
> - “Teacher” = initial language model’s (LM) performance with additional context.
> - “Pre-distill Student” = the initial LM’s performance without additional context.
> - “Post-distill Student” = the distilled LM’s performance without additional contexts
>
> We should generally expect the performance to be 1 > 3 > 2, and Condition 3 > 2 is necessary to support our claims. Notice that, since we are distilling context, **whether a system is called teacher or student is NOT based on what parameters they have, but which system uses more context tokens**. For example, “teacher” and “pre-distill” has the same parameter values, while “pre-distill” and “post-distill” students do not have the same parameter values.
>
> **Our Original Draft Compared Context Distilled models with the Critical Baselines You Recommended**
>
> As you said, if the teacher is instruction-tuned, it is only a fair comparison if the pre-distilled model is instruction-tuned. Indeed it is, and we strictly followed this protocol in our paper – as is now explicitly pointed out in the “baselines and oracles” section, **the pre-distilled student and the teacher have the same parameter values**; so if the teacher is instruction-tuned, the student’s initialization is instruction tuned as well.
>
> In Section 3.1 context distillation task, the student needs to directly return an output without using the task instruction; in other words, the task instruction needs to be internalized into the student parameters. We should expect the student to have near random performance pre-distillation since it has 0 information about what tasks to perform.
>
> > What does “Direct Gradient Descent” mean here?
>
> Given a few input-output examples for a task and a language model (or an instruction-tuned version), there are three ways to learn this task:
> - Directly fine-tune the language model on the examples with gradient descent
> - In-context learning
> - Context distillation, where we first perform in-context learning, and then distill.
>
> Here “direct gradient descent” does not mean whether the language model has been previously instruction-tuned – both the student and the teacher start with the same parameter anyways. Instead, it refers to directly fine-tuning with those examples.
>
> > Overclaim due to the use of unlabeled data?
>
> As described in Sections 3.1 & 3.2, **we did not use any unlabeled data from the test distribution**.
> - For experiments in Section 3.1, we ask Tk-instruct to generate a task input given a task description.
> - For experiments in Section 3.2, we prompt incoder with the concrete examples “database schema + [Input1, output1][input2, output2]”, and sample input3 from the LM.
>
> Section 3.3 is a synthetic experiment where we used the test distribution to synthesize data, and we have tuned down the claim for that section in the updated draft.
>
> > Motivation to choose the tasks where [natural language explanation + task description] outperforms/underperforms [task description alone] for the teacher model:
>
> To clarify, chose tasks based on whether additional natural language explanations are helpful beyond task description alone. We chose these tasks because they also allow us to investigate how well context distillation can internalize natural language explanations in Appendix 8.2 under different teacher’s performance
>
> We hope that these clarifications help!

---

> > ### Comment · Reviewer_Krod · 2022-12-04
> > **Thank you for clarifying**
> >
> > Thank you for clarifying.
> >
> > RE baseline: I did not realize the pre-distil student model is the instruction-tuned model but is tested without an instruction. The paper (and your comment) refers it as an "initial LM". Is my understanding correct that an initial LM isn't the model that is only trained on an LM objective, but is a model followed by instruction-tuning? I think that caused the miscommunication, since, an initial LM seems to indicate the model that is not yet instruction-tuned.
> >
> > Also, I think then a missing baseline is the model that is fine-tuned without instruction (or the raw LM before being instruction-tuned) and is tested without instruction. I did not mention this baseline earlier because I thought a "pre-distil student" refers to this baseline. To my mind, this is a more naive baseline than a pre-distil student based on instruction-tuned model, because it's typically a default choice to make training and testing the same as much as possible.
> >
> > RE no use of unlabeled data: I didn't realize unlabeled data is only used for one of experiments but not in others, because Section 2.1 seemed to indicate it is using unlabeled data (which doesn't seem to be intended). Just curious, which section in the new version of the paper clarifies the creation of inputs for each experiment, which you mentioned in the comment?
> >
> > I appreciate authors clarifying my confusion, but my core concerns still remain.
> > (1) The concept of "internalizing" the task is slightly misleading and is not technical to my point of view (it's more accurate to say the problem setup is "performing a new task without an instruction/in-context examples as good as it does with an instruction/in-context examples").
> > (2) The clarity of the paper needs improvements.
> > (3) Experiments are not well-organized and are inconsistent.
> > In particular, (2) and (3) are pointed by all of four reviewers.

---

> > > ### Author Response · Authors · 2022-12-05
> > > **Your feedback is really helpful; curious about further feedback on the definition of internalization**
> > >
> > > > Re internalizing performance gains:
> > >
> > > Thanks for your feedback. Indeed, the most accurate description of our intuition is: "performing a new task without an instruction/in-context examples as good as it does with an instruction/in-context examples." However, this is a cumbersome phrase to convey the underlying intuition in research communications. Do you have any other definition of "internalization" with properties not captured by the full accurate definition above?
> > >
> > > For example, I could imagine a more stringent definition of internalization might require the internal structure of the parameters to correspond to the instructions; in that case, we might re-name "internalize" to "behaviorally internalize" -- the post-distillation student is behaviorally the same as the teacher with the context.
> > >
> > > > Re initial LM:
> > >
> > > Thanks for explaining where the confusion comes from here: it does look like we used confusing terminology. We will update the name for the next version.
> > >
> > > > Re-using unlabeled data
> > >
> > > I added a sentence now in Section 2.1.
> > >
> > > "
> > > Except for the synthetic experiments in Section 3.3, all experiments in our paper create D via few-shot prompting and do not assume access to a pool of unlabeled data.
> > > "
> > >
> > > > Re other baselines:
> > >
> > > I agree that they are all useful statistics in the paper; however, I will not call them "baseline", for the following reason:
> > >
> > > - For raw LM with instructions: taken to the extreme, if we instruction-tune the LM for 0 steps to create an "instruction-tuned" initial LM, then it becomes the oracle teacher's performance rather than the baseline that our method needs to beat.
> > > - Fine-tuning without instruction: taken to the extreme, suppose we have infinite fine-tuning data and perfect optimization, then by universal approximation theorem, we should have Bayes optimal accuracy, which is the upper bound of any possible model performance.
> > >
> > > Notice that both of these arguments rely on the "taken-to-the-extreme" argument. While I do not think that they will be realized in practice, these arguments reflect that the two experimental setups you recommended highly depends on other factors that our method cannot control.

---

### Official Review · Reviewer_ipLo · 2022-10-24

**Confidence:** 3
**Correctness:** 3
**Technical Novelty And Significance:** 2
**Empirical Novelty And Significance:** 3
**Recommendation:** 5

**Clarity, Quality, Novelty And Reproducibility:**

* The authors do not clearly describe the learning objective. Section 2.4 says ‘we minimize the token-level KL divergence’, but there is no mathematical description of how they do this.
* I couldn’t follow the motivation for ‘Sequential Distillation’ in section 2.3 and the problem addressed as well as the solution described are vague.
* The evaluation setting in 3.1 is vaguely described. ‘We select 5 tasks where teacher can most significantly improve…’ - I don’t understand the motivation behind these choices.
* ‘We plot the margins for each task in Figure 5’ - The discussion relies on a figure in the Appendix. The main text should be self-contained and the reader should not have to rely on the appendix to understand the details.
* The main experimental results are discussed in Table 2 and Table 3.
  * Table 2 shows the advantage of context distillation compared to a fine-tuning baseline on a text-to-SQL task. The student performance is much worse than the teacher in this case.
  * Table 3 shows the benefit of distilling scratch-pad on an 8 digit addition task where the student performs comparable to the teacher.
From these results alone I am not convinced that this is a general method that can work well across a variety of tasks.


**Strength And Weaknesses:**

Pros
* The paper considers an interesting problem - Internalizing the performance gains from methods known to improve few-shot learning performance, but without paying the additional price of having to process a long context.
* An interesting solution is proposed.
* Experiments show some evidence of the advantage of the proposed approach.

Cons
* Technical details are missing/not clearly described.
* Experimental results are sparse and not fully convincing.


**Summary Of The Paper:**

This work addresses some of the shortcomings of few-shot learning paradigms with large language models such as learning from instructions and explanations, scratch-pad/chain-of-though reasoning and learning from demonstrations. The authors point out that the following issues are prevalent with these methods - (i) The context window can be long, which results in inference being computationally expensive (ii) The inability to fit more examples than the context window allows and (iii) The gains disappear after the demonstrations or scratchpad disappear, i.e., the model doesn’t internalize these gains. This work proposes ‘context distillation’ as a solution. A student model is trained to match the distribution of outputs of a teacher model, where the student model uses a simpler and shorter context, such as only a simple instruction without examples/explanations/scratch-pad. Experiments attempt to show the advantage of the proposed method under the different few-shot learning paradigms described above.

**Summary Of The Review:**

The authors propose a neat idea which is applicable across a variety of learning paradigms with large language models. However, the experimental results are not comprehensive enough to understand the benefit of the proposed approach.

---

> ### Author Response · Authors · 2022-11-17
> **Response to Reviewer ipLo**
>
> > Further Explanations for Token-Level KL Divergence
>
> Thanks for bringing this up. We realized this is a common confusion among the readers, so we added more explanation in Appendix 8.1 in the updated draft; we consider it to be a low-level implementation detail, and the main paper is still self-contained without it.
>
> Here’s a more intuitive description. First, we want to learn from the soft label, that includes the logit information for each vocab, rather than learning from the hard label, which is one single sample from the soft label distribution.
>
> Second, soft labels (p*) are distributions over a large vocab space, which consumes 200KB for each time step; this would either create a large memory usage on TPU/GPU, or a large communication cost. Therefore, we want to “compress” the most important information of the soft label distribution (p*) over all the vocabs.
>
> We did this by sampling 100 tokens from p*. As a toy example with 2 vocabs, if p* = [0.1, 0.9], then we create a list of samples samples_l = [1, 1, 1, 1, 0, 1, 1, ….], and \hat{p} would be the empirical distribution of the list of samples, e.g., {0: 8/100, 1: 92/100}. As the vocab size increases, storing a short list of samples is more memory efficient than storing the entire probability vectors; additionally, it is an unbiased approximation of p*. (Note that one could trim the vocab probability distribution to only the top-K vocabs; however, this is not an unbiased approximation.)
>
> > Student performance is worse than the teacher for Table 2
>
> As described in the “baselines and oracle” paragraph in Section 3 in the updated version, since we aim to internalize the benefit of using context, the teacher that uses in-context examples is an oracle for the performance, and the proper baseline is the student model before distillation without using context. Therefore, it is expected that post-distillation student is worse than the teacher with in-context examples.
>
> Finally, the gap significantly decreases when we use more in-context examples.
>
> > Tuning down the claim for internalizing step-by-step reasoning
>
> We agree that the experiments on 8-digit addition are a bit thin to support the overall claim on internalizing step-by-step reasoning, so we tuned down this claim as “preliminary evidence” and hope that this result can generalize to more complex tasks and larger models.
>
> > ‘We plot the margins for each task in Figure 5’ - The discussion relies on a figure in the Appendix.
>
> We now moved all the claims related to natural language explanations to the appendix so that the main paper is self-contained.
>
> > Motivation to choose the tasks where [natural language explanation + task description] outperforms/underperforms [task description alone] for the teacher model:
>
> We chose these tasks because they also allow us to investigate how well context distillation can internalize natural language explanations in Appendix 8.2 under different teacher’s performance

---

> > ### Comment · Reviewer_ipLo · 2022-11-23
> > **Thank you for the response**
> >
> > Thank you for the response and updates. I wish the lead author a speedy recovery.
> >
> > Regarding the KL Divergence loss, I appreciate the additional details. But I do not agree that the main paper is self-contained without it because the implementation details matter. I do think the technical details need to be discussed in the main text.
> >
> > Overall I do think the idea is interesting. But the paper can benefit from more comprehensive evaluation with clear motivations for the experimental setup and calibrating the claims (as pointed out by other reviewers as well).

---

### Official Review · Reviewer_7ryR · 2022-10-25

**Confidence:** 2
**Correctness:** 3
**Technical Novelty And Significance:** 3
**Empirical Novelty And Significance:** 3
**Recommendation:** 6

**Clarity, Quality, Novelty And Reproducibility:**

The paper overall is clear, and the quality of writing is quite good. Nevertheless, I consider the location of figures to be confusing. For example, figure 1 is referred after figure 2, while actually both figures could be almost merged into a single one, that explains the idea of template, etc., through the example of number addition. Then, while hypothesis 2 uses figure 5, this has been moved to the appendixes. Besides, figure 5 would benefit from square-axis.

For the sake of reproducibility and further supporting the statements, the authors provide further results in the appendix, as well as the code.

Finally, I found a couple typos:
- Cognative: last line 1st page
- By: in capital letters, in Table 4 caption


**Strength And Weaknesses:**

The authors propose a really interesting idea, where distillation is used to infuse a pre-trained network with further context that is associated with diverse downstream tasks, boosting their performance. The approach considered, where different hypotheses are laid out and assessed with different tests, is interesting, and helps grounding step by step the possibilities of the methodology. Most of the results are clearly presented, and successfully justify the hypothesis postulated, even though further tests on other downstream tasks would be necessary to really confirm them.

Still, I believe the authors do not tackle a really important point, critical to assess the full potential of the methodology: how the knowledge distilled in the network vanishes, or not, as it is trained on more and more tasks. Therefore, it would have been interesting seeing results of this phenomena, as another hypothesis laid out on this premise. I believe such hypothesis would have been more relevant to evaluate than some of the ones considered by the authors. The concern is that the knowledge distilled vanishes as the network is trained on more tasks, which could be indicated by the decrease in performance of those tasks. If this is the case, the demanding fine-tune required would be less justifiable, and more ad-hoc networks or other fine-tuning methods might be a better alternative.

Besides, it would be valuable to have confidence intervals in the reported accuracies, in order to understand how the distillation process can be affected by the randomness in the set of provided inputs, as most experiments are just trained on a few epochs.

Finally, I have two other questions:
- The explanation given in Section 2.4, for the simplification done on the soft-labels, is not really clear to me. Could you further elaborate?
- In table 1, my intuition tells me that, when training on a single task, i.e. naive post-distill, the performance should be better than when mixing several. Is not that right? Or is it the case that the single task for naive training is the most complicated one, among all the ones used in the Mixed training? However, according to table 6, the performance of all tasks increases in the mixed version. Could you provide some further intuition on this?

**Summary Of The Paper:**

The authors present an alternative paradigm to prompting: context distillation. The idea is that the model, instead of utilizing examples and specific prompts just for predicting without learning anything from these, tries to internalize this information. The authors propose a teacher-student architecture, where both start from the same parameters' state, and the updates of the student network are driven by the differences in x and y, fed and output respectively, by each network. While the teacher network receives a complete input, with examples and explanations, and outputs the target plus scratch-pad text, the student network generally receives some simple task explanation and the raw input, and is trained to output the raw target.

Throughout the paper, the authors come up with several hypotheses, aiming at proving these contextualization capabilities, and perform different tests to assess the correctness of these hypotheses. In all cases the student network, after distillation, outperforms the original student network, proving the potential of this approach for incorporating valid contextual information into the networks. The authors test the presented methodology in three different contexts, which requires different datasets and downstream tasks.

**Summary Of The Review:**

Overall, I believe the paper presents really interesting ideas, as well as tests and results to support the main claims. However, I still feel that the concept of knowledge vanishing has not been tackled, and it would be important to understand how it works in order to really provide a methodology that can perform in multi-task environments. I believe the authors should further discuss this, in order to improve the overall quality.

---

> ### Author Response · Authors · 2022-11-17
> **Response to reviewer 7ryR**
>
> > Updated experiments to address the concern that knowledge might be vanishing after context distillation.
>
> Thanks for bringing this up – this is indeed a crucial aspect if we want to apply context distillation iteratively to update an LM. To investigate this, we explored the recursive distillation formulation in Section 2.4 (replacing the sequential distillation experiments in our original submission).
>
> In a nutshell, we apply context distillation to an LM, and then treat the student LM as the new teacher LM for the next round of context distillation. In other words, the model needs to update itself iteratively; if it forgets how to use the context information after context distillation, this recursive algorithm will not work.
>
> We experimented with recursive distillation in the task-id association setup (Section 3.1) – the LM needs to teach itself to associate Id with task instructions via context distillation and do so repeatedly for new task Task-Id associations via context distillation. This requires the LM to know still how to use the instruction in context after previous rounds of context distillation. As shown in Table 1 right, the performance does not drop after 3 rounds of recursive distillation, indicating that its capability to follow instructions has not decreased. We conjecture that this will be a diminishing issue as model size increases [1].
>
> Finally, even though we have not encountered the knowledge vanishing issue empirically, we agree that it is a valid concern for context distillation in other scenarios. In that case, we think it’d be helpful to add instruction-tuning training data when performing context distillation to maintain its original capability.
>
> > Further Explanations for Token-Level KL Divergence
>
> We added more explanation in Appendix 8.1 in the updated draft. We consider it to be low-level implementation detail, and the main paper is still self-contained without it.
>
> Here’s a more intuitive description. First, we want to learn from the soft label, which includes the logit information for each vocab, rather than learning from the hard label, which is one single sample from the soft label distribution.
>
> Second, soft labels (p*) are distributions over a large vocab space, which consumes 200KB for each time step; this would either create a large memory usage on TPU/GPU, or a large communication cost. Therefore, we want to “compress” the most important information of the soft label distribution (p*) over all the vocabs.
>
> We did this by sampling 100 tokens from p*. As a toy example with 2 vocabs, if p* = [0.1, 0.9], then we create a list of samples samples_l = [1, 1, 1, 1, 0, 1, 1, ….], and \hat{p} would be the empirical distribution of the list of samples, e.g., {0: 8/100, 1: 92/100}. As the vocab size increases, storing a short list of samples is more memory efficient than storing the entire probability vectors; additionally, it is an unbiased approximation of p*. (Note that one could trim the vocab probability distribution to only the top-K vocabs; however, this is not an unbiased approximation.)
>
> > The performance difference between naive distillation and mixed distillation
>
> In the updated draft, we create a 95% confidence interval for Table 1. The difference between the naive distillation and mixed distillation is not statistically significant on the correct association metric. However, their difference is significant for the wrong association metric (lower better).

---

> > ### Comment · Reviewer_7ryR · 2022-12-05
> > **Thank you for the response**
> >
> > I would like to thank the authors for the clarifications. Even though hypothesis 3 partly evaluates what I was referring to on my comment, I don't think it fully reports what I meant. First, it provides the performance for all the tasks together, but as far as I understand, there is no reference to the performance of each individual task, in order to assess if it degrades as more and more tasks are added. I would have seen valuable a plot showing results for the model as we add tasks, showing the performance for the task individually evaluated, and for the model trained sequentially. Besides, this evaluation should have been made for more than 4 tasks, to assess the limits of the model.
> >
> > Besides this, I am satisfied with the provided answers, and for the detailed responses. I also would like to send all my best wishes to the main author, wishing her or him a prompt recovery.

---

### Official Review · Reviewer_fidG · 2022-10-25

**Confidence:** 4
**Correctness:** 3
**Technical Novelty And Significance:** 3
**Empirical Novelty And Significance:** 3
**Recommendation:** 5

**Clarity, Quality, Novelty And Reproducibility:**

Even though the concept of “context distillation” is not new, the authors did a good job of conducting comprehensive research on three different perspectives. However, the presentation/clarity of the paper can be significantly improved, as suggested in the “weakness” section. Besides, I have several questions for the paper:

1. (Sorry if I missed it) what model did you use to generate the input and how many inputs did you generate for each task?
2. I would be curious to see some examples of generated input.
3. Is it possible to provide more results in exp#2 on datasets other than SQL?
4. Have you thought about/tried other optimization methods other than fine-tuning? E.g., soft prompt tuning?
5. For the step-by-step reasoning experiment, have you tried it on models that are not fine-tuned on NatIns? E.g., directly using chain-of-thought on an LLM?


**Strength And Weaknesses:**

# Strength

The authors conduct a systematic study of “context distillation” in three aspects: instructions, demonstrations, and step-by-step reasoning. The method is intuitive and the results are strong. It is not only a way to improve the efficiency of using in-context learning and instruction prompting, but the fact that the knowledge can be “internalized” is very interesting.

# Weakness

The experiment setup can be significantly improved. First, the paper can be improved on how they demonstrate the results. There are very few tables/figures on the results in the main paper and most numbers scatter in the text. Second, the clarity of the experiment setup (what datasets they use for exp#1, how many pseudo input examples they generate, etc. sorry if I missed it) can be improved. For Exp#2, it is very limited to only test context distillation on the SQL dataset. It would be better to show other datasets too, like NLU, QA, etc.

**Summary Of The Paper:**

This paper proposes “context distillation”, a method to “internalize” the information provided by abstract instructions, concrete demonstration examples, or scratchpad (model intermediate output that helps reasoning). They first use a language model to sample some input (e.g., movie reviews), and then they use the teacher model to generate pseudo answers given inputs as well as demos, instructions and generated scratch-pad. The pseudo input-answer pairs are used to fine-tune the student model (minimizing the KL divergence using teacher logits), without the extra information. The benefit of internalizing the knowledge is that there is no longer a need to use the extra input and output, making the use of LLM more efficient. The difference to traditional distillation is that the teacher and student models are the same one, and it is the “extra information” that gets distilled into the student model. The authors conducted three experiments:

Exp#1: learning from instructions. They first fine-tune the model on Natural-Instruction-V2 (a T5-11B). They show that they can significantly improve the student model’s performance without instruction or explanation.

Exp#2: learning from examples. The authors use SPIDER text-to-SQL as a test bed. They show that context distillation can effectively internalize the examples in the context and outperform direct fine-tuning.

Exp#3: learning from step-by-step reasoning. The authors test the model on 8-digit addition questions. The experiment shows that the student model can even internalize the step-by-step reasoning by fine-tuning on teacher model’s output, without using scratchpad.

Even though the concept of “context distillation” was already proposed by Choi et al.; Askell et al., this paper did a good job of systematically testing it on three distinctive aspects, and the experiment results show that context distillation is an effective way to “internalize” knowledge from the “context”.


**Summary Of The Review:**

Overall the paper is well written and the conducted research is insightful and the results are strong. The paper should be accepted. The quality of the presentation can be further improved per the weakness section.

---

> ### Author Response · Authors · 2022-11-17
> **Response to reviewer fidG**
>
> > (Sorry if I missed it) what model did you use to generate the input and how many inputs did you generate for each task?
>
> I will double-check with the 1st author when he can reply to my message. If I recall correctly, for each task, we generated 1024 inputs for each teacher template (since we are simultaneously distilling 4 tasks in Section 3.1 for hypothesis 2, there are 4096 inputs in total) and trained the student for 1 epoch; we generated them with few-shot prompting on the teacher model with temperature 1.
> - For experiments in Section 3.1, we ask Tk-instruct to generate a task input given a task description.
> - For experiments in Section 3.2, we prompt incoder with “database schema + [Input1, output1][input2, output2]”, and sample continuations from the LM, which we expect to be another input.
>
> > I would be curious to see some examples of generated input.
>
> We will include this in the appendix as soon as the 1st author recovers from the concussion.
>
> > Is it possible to provide more results in exp#2 on datasets other than SQL?
>
> Thanks, we will provide results for more tasks from the evaluation set of Natural-Instruction-V2 using Tk-Instruct (based on T5-11B). For many of those results, I’d predict that In-context learning performance \approx gradient descent > context distillation > 0-shot learning. In other words, context distillation will be successful compared to no distillation but might be worse than direct gradient descent.
>
> We picked Text-to-SQL because we know that it benefits greatly from in-context examples even for models with as few as 6B parameters. In contrast, the performance gain of in-context learning over gradient descent for many other tasks is dubious for model sizes we can fine-tune. We expect in-context learning to work better when we further scale up.
>
> Note that such a design choice does not count as cherry-picking since we choose tasks based on their in-context learning performance rather than the context distillation performance. This is consistent with our motivation: context distillation aims to internalize the benefits of context information, so we shouldn’t expect it to work well if the LM cannot effectively leverage the in-context examples.
>
> > Have you thought about/tried other optimization methods other than fine-tuning? E.g., soft prompt tuning?
>
> Thanks for the recommendation. We have not thought about this, which would be an interesting future direction. We consider the optimization method to be a complementary direction to using context distillation.
>
> Additionally, in our new recursive distillation setup, we want the model to iteratively update itself without introducing new parameters. Therefore, soft prompts might not be the best choice here since it introduces new (though very few) parameters. Fine-tuning the bias term or the layer norm could be a more suitable choice here.
>
> > For the step-by-step reasoning experiment, have you tried it on models that are not fine-tuned on NatIns? E.g., directly using chain-of-thought on an LLM?
>
> Thanks for bringing this up. Most tasks only benefit marginally from CoT for the model sizes we can fine-tune, so we have not experimented with them yet; additionally, it is unclear whether those gains come from the use of intermediate reasoning steps rather than “distribution alignment” – where a more logically coherent prompt is more likely to elicit correct answer on an LM that is not instruction-tuned.  Working with an instruction-tuned LM and 8-digit addition allows us to rule out the alternative hypotheses cleanly.
>
> Nevertheless, we agree with ipLo that the experiments on 8-digit addition are a bit thin to support the overall claim on internalizing step-by-step reasoning, so we tuned down this claim as “preliminary evidence” and hope that this result can generalize to more complex tasks and larger models.

---

### Author Response · Authors · 2022-11-17
**General Response to Reviewers**

Sorry for the slow response – I’m the last author of this submission and would like to thank all reviewers for their response. The 1st author is still recovering from a serious concussion, so our response will be brief, and we cannot run many of the recommended experiments. We do not expect the AC and the reviewers to lower their standards. Still, we want to clarify that the brevity is because of unforeseen circumstances rather than us not appreciating the reviewers’ time and commitment. We are grateful for the reviewers’ effort to read through the paper carefully and provide insightful comments.

We thank reviewers for appreciating the novelty of the problem we are tackling and the writing quality. We updated the draft according to the reviewer’s recommendation and included new experiments we ran after the ICLR deadline before the response period, which addresses some of the reviewer’s concerns. For experiments that we are unable to run, we will openly discuss our best prediction about the experimental results, drawing indirect evidence from this paper or conclusions from related works; some of the predictions work in our favor, and some of the others do not.

Major updates for the paper draft:
- Related to reviewer 7ryR’s concern about LM forgetting its previous skills after context distillation, we included new experiments on recursive distillation, where the LM can iteratively update itself to associate ID with tasks, without losing its ability to follow the instructions.
- To address reviewer 7ryR and ipLo’s question about token-level KL divergence, we added a figure and more explanations in Appendix 8.1.
- As reviewer 7ryR recommended, merged figures 1 and 2.
- Based on feedbacks from reviewers ipLo and fidG, we tuned down the claim on step-by-step reasoning. 8-digit addition is a proof-of-concept. We hope that our results can generalize to more complex tasks, which, so far, can only be performed by much larger language models that we cannot fine-tune.
- Moved the experiments on learning from natural language explanations (original hypothesis 2) to the appendix due to space constraints.
To clarify potential misunderstandings by reviewer Krod and  ipLo, we added more signposting at the beginning of Section 3.1 to discuss the design choice and what to expect for the experiments.

---

### Decision · Program_Chairs · 2023-01-20

**Decision:**

Reject

**Justification For Why Not Higher Score:**

The paper is decent but not sufficiently convincing in its experiments so acceptance was not on the table.

**Justification For Why Not Lower Score:**

N/A

**Metareview: Summary, Strengths And Weaknesses:**

The authors present a method to distill (fine-tune) a language model, trained using context information in the form of abstract instructions, demonstrations and the use of a scratchpad, into a model which performs almost as well on a variety of tasks using without the context information.  This is an important problem in language model prompting which otherwise suffers test-time inference costs and memory limitations from the context prompting requirement.

Roughly the procedure is to:
1. sample data from a task of interest and use a teacher LLM to generate pseudo-answers
2. Fine tune a student model to predict the same answers as in (1) but without the context.

There are 3 main experiments: (1) learning from instructions; (2) learning from examples; (3) learning from step-by-step reasoning.

All reviewers felt that the paper makes a good contribution on a practical problem and we appreciated the author’s clarifications (made more challenging by an unfortunate health issue for the first author).  After discussion, we all believe that the paper could be made stronger in two core respects:

1. it could use more clarity in the experiment description and details (see individual reviews for specifics).
2. it could use a more comprehensive evaluation on additional tasks. For example, experiment 2 which uses Text-to-SQL could be run on a broader range of NLP datasets which include NLU, QA, etc.  The authors argue that they chose the datasets based on their ability to improve from context prompting, which is reasonable but also begs the question of how broadly useful prompting is and therefore how impactful this technique will be. There is surely room for additional experimentation on the 8-digit addition experiments in any case.

In its current state the reviewers felt that it was a little too underbaked to accept but we all liked the idea and encourage the authors to address these issues and resubmit.


**Summary Of Ac-Reviewer Meeting:**

The paper started off borderline but after a short discussion by email it was clear it was not going to be accepted so we just had a short email discussion among the reviewers which was entirely sufficient.

All reviewers felt similarly that the idea of the paper was clear and reasonable but the experiments were not well-explained or convincing. The initial spread of reviewer scores reflected a difference between them in how strongly to respond to the deficits, but all agreed they were there.  After a short discussion we agreed to settle on lower scores and reject.